# Enniatin B and Deoxynivalenol Activity on Bread Wheat and on *Fusarium* Species Development

**DOI:** 10.3390/toxins13100728

**Published:** 2021-10-15

**Authors:** Luisa Ederli, Giovanni Beccari, Francesco Tini, Irene Bergamini, Ilaria Bellezza, Roberto Romani, Lorenzo Covarelli

**Affiliations:** 1Department of Agricultural, Food and Environmental Sciences, University of Perugia, 06121 Perugia, Italy; luisa.ederli@unipg.it (L.E.); francesco.tini@collaboratori.unipg.it (F.T.); irenebergamini95@gmail.com (I.B.); roberto.romani@unipg.it (R.R.); lorenzo.covarelli@unipg.it (L.C.); 2Department of Medicine and Surgery, University of Perugia, 06132 Perugia, Italy; ilaria.bellezza@unipg.it

**Keywords:** wheat, deoxynivalenol, enniatins, mycotoxins, *Fusarium*, cell death

## Abstract

*Fusarium* head blight (FHB) is a devastating wheat disease, mainly caused by *Fusarium graminearum* (FG)—a deoxynivalenol (DON)-producing species. However, *Fusarium avenaceum* (FA), able to biosynthesize enniatins (ENNs), has recently increased its relevance worldwide, often in co-occurrence with FG. While DON is a well-known mycotoxin, ENN activity, also in association with DON, is poorly understood. This study aims to explore enniatin B (ENB) activity, alone or combined with DON, on bread wheat and on *Fusarium* development. Pure ENB, DON, and ENB+DON (10 mg kg^−1^) were used to assess the impacts on seed germination, seedling growth, cell death induction (trypan blue staining), chlorophyll content, and oxidative stress induction (malondialdehyde quantification). The effect on FG and FA growth was tested using ENB, DON, and ENB+DON (10, 50, and 100 mg kg^−1^). Synergistic activity in the reduction of seed germination, growth, and chlorophyll degradation was observed. Conversely, antagonistic interaction in cell death and oxidative stress induction was found, with DON counteracting cellular stress produced by ENB. *Fusarium* species responded to mycotoxins in opposite directions. ENB inhibited FG development, while DON promoted FA growth. These results highlight the potential role of ENB in cell death control, as well as in fungal competition.

## 1. Introduction

The genus *Fusarium* is distributed worldwide, with at least 300 phylogenetically distinct species [1]. The pathogenic species are generally considered to be hemibiotrophic, able to switch—after an initial and short biotrophic phase—to necrotrophy [2,3]. *Fusarium* species are able to cause diseases in many crops, and they can be particularly harmful to wheat and barley [4]. In these crops, several *Fusarium* species cause *Fusarium* head blight (FHB)—a devastating disease resulting in yield and quality losses [5]. Some *Fusarium* species are also opportunistic pathogens to humans, causing infection to various tissues, such as the cornea and nails [6].

The incidence of each species within the FHB complex can vary depending on the type of cereal, geographic region, and environmental conditions that occur especially at anthesis, the highest susceptibility stage of wheat [7]. Among the various species involved in FHB, *Fusarium graminearum* (FG) is considered the most prevalent worldwide [8]. However, in recent years, another species, *Fusarium avenaceum* (FA), has been detected in wheat and barley grain with an increasing incidence in many cultivation areas across the world [9,10]. FA is a fungal pathogen that causes various diseases in many crops, such as root and stem rot in various host species [11,12,13], as well as FHB in cereals [14]. In some Northern European countries, FA has long been considered one of the dominant species in the *Fusarium* community associated with FHB [14,15,16,17]. However, in other cultivation areas throughout the world—including warmer regions—FA was also reported as the most frequently detected species in wheat grains [18,19,20,21]. In particular, during the past decade, this species has also increased its frequency in Italian wheat and barley, often showing co-occurrence with FG [22,23,24,25,26].

One of the peculiarities of the *Fusarium* spp. associated with FHB is represented by their ability to biosynthesize different mycotoxins. These compounds can contaminate agricultural raw materials, starting from the preharvest phase up to the postharvest [27]. Within the whole range of these secondary metabolites, the trichothecene deoxynivalenol (DON)—mainly produced by FG—is considered one of the major mycotoxins due to its global spread and its well-known toxicological impact on animal and human health [28,29]. For this mycotoxin, a role in fungal virulence [30], as well as action towards plants [31], has also been demonstrated.

Despite the growing interest in mycotoxins, most previous studies have focused on DON, while enniatins (ENNs) have received little attention thus far [32,33]. Recently, the influence of ENNs in FA virulence in wheat was investigated [34]. However, knowledge about the effects of ENNs on wheat plants is still scarce and based on dated studies [35,36]. In addition, ENNs activity in wheat in combination with major mycotoxins such as DON is poorly explored. There is some evidence of ENNs’ effects on bacterial and fungal microorganisms [37,38]; however, the possible interaction with DON has also not been elucidated in this case.

FA is one of the most important ENNs producers [39,40], and these compounds—enniatin B (ENB) in particular—are common grain contaminants worldwide [9,33,41], including in Italy [42,43,44]. Often, a co-occurrence between ENNs and DON has been detected in the same analyzed grains [25,45]. Despite their high incidence, ENNs are not yet included in the European Union (EU) regulations for cereal products, unlike DON [46]. On the other hand, given the growing evidence of their presence in both food and feed [47], and since concerns about possible interactions with other mycotoxins and chronic exposure were highlighted [48], the EU Commission might evaluate the regulatory status of ENNs soon.

For these reasons, the importance of studying mycotoxin activity—not only as single independent molecules, but with an approach considering their increasing co-occurrence—is particularly evident. This is applicable to all mycotoxins, but especially to “emerging” ones such as ENNs, which are now also receiving increasing attention in terms of their ecological significance at the plant level. Therefore, in this study, we first evaluated the activity of ENB (one of the most frequently detected ENNs in wheat) and of the mycotoxin DON, applied individually or in combination (ENB+DON) on bread wheat. In particular, we investigated their effects on seed germination, root and shoot growth, chlorophyll level, cell death, and oxidative stress induction. In addition, we also analyzed the possible synergistic or antagonistic roles of different concentrations and combinations of these two mycotoxins on the in vitro development of both FG and FA.

## 2. Results

### 2.1. Effects of ENB and DON on Bread Wheat Germination and Seedling Growth

In order to evaluate the phytotoxic action of the two mycotoxins ENB and DON, alone or in combination, the effect produced on bread wheat germination was initially examined. After four days, germination rate (%) did not show significant changes in comparison to the untreated control as a consequence of the treatments with either ENB or DON alone (Figure 1). Conversely, the exposure of bread wheat seeds to the combination of the two mycotoxins (ENB+DON) significantly reduced (−15%) their germination rate (Figure 1).

The potential phytotoxicity of ENB, DON, and ENB+DON was also assessed on bread wheat seedlings’ growth by analyzing root and shoot development at four days after mycotoxin application. The obtained results clearly showed that bread wheat was more sensitive to DON, which caused an evident and significant decrease in both shoot (−16.25%) and root (−11.5%) growth in comparison to the untreated control (Figure 2). The treatment with ENB alone caused a reduction in shoot development comparable to that induced by DON, whereas no significant effects on root growth were detected (Figure 2). When the two mycotoxins were used in combination, their action was synergistic, resulting in a greater inhibition with respect to the untreated control, which was evident only at the root level, where ENB—which alone did not show any significant effect—significantly reduced root elongation in association with DON (Figure 2). In addition, the analysis of the seedling “vigor index” in response to ENB, DON, or ENB+DON combination mirrored the previous results. In fact, also in this case, the most consistent action of DON—which alone reduced seedling vigor in terms of both size and weight—was found compared to the untreated control (Figure 3). Indeed, the treatment with ENB alone affected only the seedling weight vigor index, but it was the combination with DON that caused the inhibitory effect already shown when applied alone on the vigor index assessed on the basis of both weight and length (Figure 3).

### 2.2. Cell Death and Oxidative Stress in Response to ENB and DON

Experiments using trypan blue staining allowed for the quantification of cell death induced by the two applied mycotoxins, alone or in combination, through vacuum infiltration of bread wheat leaf segments. Results showed a contrasting effect between ENB and DON (Figure 4). In detail, ENB (10 mg kg^−1^) strongly induced cell death, producing a staining intensity almost double (+90%) that of the control. On the other hand, DON (10 mg kg^−1^) produced a very slight non-significant reduction in cell death compared with the untreated control (Figure 4). The combination of the two mycotoxins caused a significant increase in cell death (approximately +54%) compared to the untreated control, but less evident than that produced by ENB alone (Figure 4).

The content of malondialdehyde (MDA) (an indicator of lipid peroxidation and a commonly accepted marker of oxidative stress) significantly increased, in both shoots and roots, after treatment with ENB (10 mg kg^−1^) (Figure 5). A greater increase (+54%) in shoots than that detected in roots (+29%) was observed (Figure 5). In contrast, DON (10 mg kg^−1^) did not induce significant changes in lipid peroxidation in either shoots or roots (Figure 5). Additionally, no significant changes in MDA content in comparison to the control were detected in the shoots and roots of seedlings treated with the combination of the two compounds, indicating that in DON+ENB co-application, DON inhibits the induction of lipid peroxidation caused by ENB (Figure 5).

### 2.3. Changes in Pigment Content after Treatment with ENB and DON

Leaf segments of bread wheat infiltrated with ENB, DON, or ENB+DON and incubated for 16 h under white light for seven days showed a protective action by both mycotoxins against chlorophyll degradation (Figure 6). ENB (10 mg kg^−1^) revealed the most evident effect, even if the association of the two metabolites showed the highest protection level; in fact, while in the untreated control chlorophyll appeared almost totally degraded after seven days of incubation, in the leaf samples infiltrated with an ENB+DON mixture, the pigments were almost completely non-degraded (Figure 6).

### 2.4. Effects of ENB and DON on FA and FG Growth

To assess the activity of ENB and DON on FA and FG growth, the development of the two strains in the presence of the two mycotoxins, individually and in combination, was evaluated. The effects of ENB and DON on fungal colony growth were quantified by calculating the area of the fungal colony (cm^2^) reached after 24, 48, 72, and 96 h of incubation following the incorporation of each mycotoxin at concentrations of 10, 50, and 100 mg kg^−1^ to the growth media.

Compared to the corresponding control, the treatment with DON did not induce significant changes in FG development at the lower concentrations (10 and 50 mg kg^−1^), but it significantly reduced (−30%) colony growth starting from 48 h at 100 mg kg^−1^ (Figure 7A–C). In the subsequent hours, lower fungal growth was still evident compared to the control, although the decrease in mycelium area over time was less noticeable (−25% at 72 h and −16% at 96 h; Figure 7A–C). In contrast, DON application produced a significant increase in FA colony development in comparison to the corresponding control at all concentrations tested, which was particularly evident after 72 h (Figure 7D–F).

The addition of ENB to the growth medium showed no effect on FG colony development with respect to the corresponding control at 10 mg kg^−1^ (Figure 8A); however, it caused significant inhibition of its development at higher concentrations (50 and 100 mg kg^−1^; Figure 8B,C). In detail, the difference with DON was that mycelium reduction (−20%) was significant starting from 50 mg kg^−1^ at 96 h (Figure 8B). At the concentration of 100 mg kg^−1^, the reduction in colony growth was significantly marked as early as 48 h (about −30%), remaining in the same proportions until 96 h (Figure 8C). Regarding the effects of the ENB on FA (Figure 8D–F), the positive effect on growth compared to the corresponding control seen with DON was also evident with this mycotoxin, but to a lesser extent; in fact, the increase in colony development occurred only after a treatment with the highest concentration (100 mg kg^−1^) of this mycotoxin (Figure 8F).

Simultaneous exposure of FG to both mycotoxins (ENB+DON) resulted in a general decrease in the fungal growth at all mycotoxin concentrations; however, this difference was statistically significant only at the concentration of 100 mg kg^−1^ in comparison to the relevant control (Figure 9). Surprisingly, when the growth medium was amended with the ENB+DON mixture, a significant reduction in FA development was detected at all concentrations tested, starting as early as 48 h (Figure 9D–F).

## 3. Discussion

In the present study, we evaluated the effects on bread wheat tissues of two mycotoxins, at concentrations usually found in nature, produced by causal agents of important *Fusarium* wheat diseases such as FHB, *Fusarium* crown rot (FCR), and *Fusarium* root rot (FRR). One of the mycotoxins examined was ENB—considered an “emerging” mycotoxin—while the other one was the well-known and -studied DON. Our research was conducted using pure, commercially available compounds of these two mycotoxins applied alone or in combination to highlight, in the latter case, possible synergistic or antagonistic effects.

The potential phytotoxic action of the two mycotoxins on the development of bread wheat was evaluated considering various parameters, such as seed germination rate and seedling growth. The results showed that DON was the compound with the highest negative impact on wheat seedlings’ growth. This effect was particularly evident following root application, at the level of shoot expansion. Trichothecenes, the chemical family to which DON belongs, are known for their toxic effects on eukaryotic cells, including protein and nucleic acid synthesis inhibition, mitochondrial damage, and alteration of cell division and membrane systems [49,50,51]. For example, Bandurska et al. [52] reported that treatments with DON caused an increase in free proline related to the mycotoxin concentration tested, attributing the high level of the free amino acid to its reduced incorporation into proteins. It was also shown that DON applications strongly inhibited cell division in the germinated roots of various plant species, with the most significant mitosis reduction detected in wheat and beans [53]. These effects at the cellular level, especially on the cell cycle and protein synthesis, are consistent with DON phytotoxicity, which has been shown to cause plant growth delay with, for example, the inhibition of coleoptile and wheat shoot elongation [54]. In other experiments, DON was also shown to significantly reduce root growth during germination [55]. Decreased percentages of root elongation and relative fresh weight have also been recently shown in *Arabidopsis thaliana* [56].

The emerging mycotoxin ENB mainly showed its negative effect on shoot elongation, while it did not produce changes in root development, despite these being the organs in direct contact with the mycotoxin. Concerning the phytotoxic activity of ENB, studies are limited, although it has been reported that ENNs in general can induce germination inhibition and increased plant wilting [35,36].

To the best of our knowledge, the association between ENB and DON, and the impacts on plants due to their possible synergistic or antagonistic effects, have not been investigated thus far. Our results show that when the two mycotoxins are combined, the negative impact on wheat seedlings’ growth and vigor increases, indicating a negative synergistic effect on plant development. This same trend was observed with respect to germination rate, as the two mycotoxins alone did not cause any significant impact, but they significantly reduced the percentage of germinated seeds when used in combination.

In addition, the induction of cell death in response to ENB and DON was evaluated. The data obtained clearly indicate an antagonistic action of the two mycotoxins. In fact, at the tested concentration of 10 mg kg^−1^, DON exhibited a protective effect when compared to ENB, which produced a strong intensification of cellular damage. The same response was obtained in terms of the generation of oxidative stress and consequent lipid peroxidation. MDA content increased only after ENB application, and the association with the mycotoxin DON reduced MDA levels. In plants, lipid peroxidation is induced by different types of oxidative stress caused by various factors, such as injuries, exposure to high and low temperatures, osmotic stresses, or following pathogen attack [57]. In the existing literature, contradictory studies on the role of DON on cell death are available. Some studies on wheat showed that treatments with varying concentrations of this mycotoxin (1–100 mg kg^−1^) led to the induction of oxidative stress with accumulation of hydrogen peroxide, and increased cell death in leaves [31]. Wang et al. [56] reported that the exposure of *Arabidopsis* leaves to DON caused oxidative burst, resulting in an inhibition of the plant antioxidant systems and a subsequent increase in lipid peroxidation. In contrast, another study on *Arabidopsis* cell cultures highlighted the suppression of cell death after heat treatment with low levels (10 mg kg^−1^) of DON [58], in agreement with our results. Inhibition of cell death induced by low DON concentrations (e.g., 10 mg kg^−1^) could favor the establishment of FG which, being a hemibiotrophic pathogen, has an early biotrophic phase, followed by a necrotrophic phase supported by higher levels of the mycotoxin [58]. Indeed, DON is a fungal virulence factor that facilitates FHB infections [30]. DON production typically begins 24 h after inoculation [59], with a significant increase in its concentration only after 96 h [60]. Thus, the switch from biotrophy to necrotrophy is specifically associated with an increase in the biosynthesis of DON [61,62] which, through translocation, reaches the healthy spikelets [63], facilitating the spread of the disease.

Very little is known about the effects of ENB on oxidative stress, lipid peroxidation, and cell death in plants. It was observed that ENB production by FA increased necrosis in potato tubers [64]. Recent studies showed that this mycotoxin induced a strong accumulation of reactive oxygen species (ROS), resulting in cytotoxic effects in murine embryos [65]. Exposure of immortalized human cell lines to low levels of ENB also caused alterations of mitochondrial proteins and electron-transporting membrane complexes [66], and promoted oxidative stress. Moreover, repeated applications of this mycotoxin resulted in glutathione reduction and ROS accumulation in the brain [67]. On the other hand, it is precisely its lipophilic nature in addition to its ionophoric activity that makes it potentially capable of altering membrane permeability and inducing cell death. It has already been shown that lipid peroxidation in tobacco can be induced by treatments with other ionophores [68], which can alter membrane potential and overall stability by modifying the functioning of ion channels.

In this study, the effects of the two mycotoxins on plant pigments were also examined; our results showed a positive synergistic action of ENB and DON, similar to that observed on growth. Indeed, infiltration of leaf tissues with the two mycotoxins alone (especially with ENB) already partially reduced pigment degradation, but their combination caused the most significant effect, with chlorophyll that remained stable in the light. Bushnell et al. [69] showed that, following some treatments with nontoxic concentrations of DON, leaf tissues remained green, without bleaching or other signs of injury, indicating a senescence delay compared to the slow yellowing of untreated leaf segments. This effect could always be related to its action at low doses (e.g., such as during the early infection phases) on cell death inhibition and, thus, on senescence. When the concentration of DON increases, as in the later stages of the disease, it becomes toxic and induces cell death, leading to various types of damage, including tissue bleaching.

It remains more difficult to explain the effects of ENB since, having demonstrated its ability to induce oxidative stress and lipid peroxidation, an increase in pigment degradation and no protection from yellowing were expected. However, many plant pathogens, including fungi, may use their effectors to manipulate phytohormones as an attack strategy [70]. It could be hypothesized that ENB is produced by FA to alter, for example, the biosynthesis and accumulation of phytoregulators, such as cytokinins, by delaying leaf yellowing and general senescence. This would promote the availability of nutrients to facilitate fungal development and, thus, the infection process. Cytokinins can be produced by the pathogenic fungi themselves to promote infection and disease [71]. Recently, in fact, a new class of cytokinins has been identified in *Fusarium* that would appear to be involved in suppressing the plant defense response during colonization, although its function in virulence has not yet been demonstrated [72].

Finally, the role of ENB and DON in the development of FG and FA was investigated in order to also evaluate their ecological significance in the competition between *Fusarium* spp. that can co-occur in the same plant or tissue (e.g., the head). Three different concentrations (high, medium, and low) were tested. The results of the in vitro growth experiments were particularly interesting, as the two tested *Fusarium* spp. (FG and FA) reacted to the treatments in opposite manners. Each mycotoxin, applied individually, produced the inhibition of FG development—mainly at high concentrations—while they caused an increase in FA growth even at low doses.

The ability of FG to produce DON could provide an advantage to this pathogen during competition with other eukaryotic organisms [73], due to its action as a protein synthesis inhibitor. Despite this, here we observed that DON not only did not reduce the development of FA, but it rather significantly induced its growth, highlighting a thus-far unknown action of DON, which instead of counteracting the *Fusarium* species that do not produce this mycotoxin, favored them. On the other hand, no reduction in the biomass of other competing fungi (e.g., *Trichoderma atroviride*) was observed in previous co-inoculation studies [74]. Thus, DON is a well-known key compound in FHB’s development on wheat [31], but it does not seem to be an important factor in *Fusarium* interspecific competition. In addition, the inhibition of FG growth observed in the presence of 100 mg kg^−1^ DON was probably related to the very high dose used, which is never normally found in nature.

To date, there is no clear evidence of the direct effects of pure ENB on the development of other fungi [38]. From the in vitro experiments performed in this study, we hypothesize that ENB could be a potential critical component in the competition between *Fusarium* spp. for example, negatively interfering with the development of FG, a fungus that does not produce this mycotoxin, and favoring FA, the fungus able to biosynthesize it. Thus, this mycotoxin does not appear to be a key metabolite in disease development on wheat and peas [34], but it could be an important factor in *Fusarium* interspecific competition. However, when ENB and DON were combined, a significant reduction in the growth of both *Fusarium* spp. tested in this study was observed, indicating that their co-occurrence caused strong toxicity for FG and FA. Willsey et al. [75] demonstrated that FA along with the root rot pathogen *Aphanomyces euteiches* increased disease severity in pea fields affected by root rot compared to infections caused by only one pathogen. Furthermore, in vitro experiments with mammalian cells showed that the association of ENB with other mycotoxins—e.g., DON—significantly increased their toxicity [76].

## 4. Conclusions

The aim of this work was to investigate the role of two important mycotoxins produced by two different *Fusarium* species in their phytotoxicity towards bread wheat, and in the competition between fungi occupying the same ecological niche. The two mycotoxins tested in this study were the well-known mycotoxin DON—synthesized by FG—and ENB, an “emerging” mycotoxin produced by FA.

The results obtained showed that ENB and DON—individually, but mainly in combination—have negative effects, especially on the growth and vigor of bread wheat seedlings. A positive synergistic effect was instead evidenced in terms of the protection against pigment degradation by light and subsequent yellowing. A contrasting action was observed regarding the induction of cell death and MDA accumulation. In this case, in fact, at the concentration of 10 mg kg^−1^, DON did not stimulate cell death, while ENB significantly increased it together with oxidative burst and consequent lipid peroxidation. Moreover, the combination of the two mycotoxins showed an antagonistic interaction, with DON counteracting the induction of cellular damage produced by ENB.

Experiments related to the competition between *Fusarium* species also indicated a different role of the two mycotoxins. In fact, ENB, more than DON, seemed to be ecologically important, as it inhibited FG growth, whereas it stimulated that of the producing fungus, FA. In contrast, DON showed no significant effects except at very high concentrations, which caused inhibition of the mycelium development of both of the *Fusarium* species tested. The use of single strains of FG and FA does not allow a general conclusion to be drawn at the species level; however, the present results can enhance the information on the response of these two FHB pathogens to DON and ENB mycotoxins, and to their combination.

This study, therefore, contributes to improve the knowledge on the action of these two mycotoxins on both the plant and the fungus, providing, in particular, indications about their synergistic or antagonistic effects. Not all possible functions of mycotoxins have yet been defined, and the study of the overall role of these compounds in natural habitats is currently of increasing interest in the scientific community. Our results especially highlight the potential role of ENB in wheat cell death promotion, as well as its involvement in the specific interaction with FG, where it appears to offer an ecological advantage in interspecific competition. In the context of toxigenic fungal pathogens, these studies are important to provide information on the co-occurrence of mycotoxins that, also as a result of the increasingly evident climate changes, could be very common in nature and have an impact on food safety and security.

## 5. Materials and Methods

### 5.1. Plant Material and Mycotoxins

Bread wheat (cv. A416, with well-known susceptibility to *Fusarium* spp.) was used for all experiments. The pure compounds ENB (BioAustralis Fine Chemicals, Smithfield, Australia) and DON (Sigma-Aldrich, St. Louis, MO, USA) were used for all experiments. Both mycotoxins were solubilized in dimethyl sulfoxide (DMSO) (Sigma-Aldrich), aliquoted, and stored at −20 °C until use.

### 5.2. Germination Assay

Bread wheat seeds (cv. A416) were surface sterilized with 1% sodium hypochlorite for 10 min, rinsed five times with sterile water, and placed in the dark at 4 °C for three days. Subsequently, the seeds were incubated for 24 h in a solution of DON (10 mg kg^−1^), ENB (10 mg kg^−1^), or a combination of the two mycotoxins (DON 10 mg kg^−1^ + ENB 10 mg kg^−1^) under gentle agitation, and then placed in 10 cm Petri dishes (10 seeds per plate) on two layers of filter paper to which 5 mL of sterile water was previously added. The Petri dishes were moved to a growth chamber maintained at 22 ± 2 °C, under 16/8 h day/night periods with 150 μmol m^−2^ s^−1^ illumination. The germination rate (%) was quantified after four days in four separate experiments, with 10 replicates per treatment for each experiment. A seed was considered to have germinated when the radicle emerged by 2 mm in length. In all experiments, for each test, the same volume and concentration of DMSO was used as a control.

### 5.3. Seedlings Growth Analysis

Bread wheat seeds (cv. A416) were sterilized as previously described and, after three days at 4 °C in the dark, they were pre-germinated in 10 cm Petri dishes (10 seeds per plate) on two layers of filter paper to which 5 mL of sterile water was previously added, and kept for two days at 22 °C. Subsequently, pre-germinated seeds were incubated for 24 h in the presence of DON (10 mg kg^−1^), ENB (10 mg kg^−1^), or a combination of the two mycotoxins (DON 10 mg kg^−1^ + ENB 10 mg kg^−1^) in 50 mL plastic tubes. Wheat shoot and root elongation, in addition to fresh weight, were assessed after four days of growth on 1% water agar media in a growth chamber with the above conditions. Four independent experiments were carried out, with ten replicates per treatment for each experiment. In addition to shoot and root elongation, wheat seedling length and weight vigor index were also calculated, as follows:seedling length vigor index = (mean of shoot length + mean of root length) × percentage of seed germination.
seedling weight vigor index = mean of seedling fresh weight × percentage of seed germination.

### 5.4. Cell Death Assessment and Pigment Visualization

In order to evaluate the effects of the two mycotoxins on plant cell death and pigment contents, leaf segments taken from three-week-old bread wheat plants (cv. A416) were entirely vacuum infiltrated with DON (10 mg kg^−1^), ENB (10 mg kg^−1^), or a combination of the two mycotoxins (DON 10 mg kg^−1^ + ENB 10 mg kg^−1^) using a syringe without a needle. Samples were then placed in 4 cm Petri dishes on a layer of filter paper moistened with treatment solutions (4 mL), and kept under gentle agitation in the growth chamber, maintaining the above conditions of temperature, photoperiod, and illumination. In all experiments, for each test, the same volume and concentration of DMSO was used as a control. Analysis of cell death was performed by trypan blue staining after seven days. Briefly, leaf segments were stained with trypan blue solution (0.01% trypan blue (*v*/*v*) in lactic acid: phenols: glycerol: water (1:1:1:1 *v*/*v*)) for 1 h, and then clarified in 96% (*v*/*v*) ethanol in a 60 °C water bath for 15 min. Samples were rinsed three times (5 min each) with deionized water and observed. Cell death was quantified using ImageJ image analysis software [77]. Leaf pigment visualization was directly performed seven days after the treatment with mycotoxins. Three independent experiments were carried out, with 8–10 leaf segments/treatment for each experiment.

### 5.5. MDA Content

Lipid peroxidation was estimated via MDA formation, assayed using a thiobarbituric acid (TBA) method. Roots of seven-day-old wheat seedlings (cv. A416) grown on 1% water agar medium were incubated for 48 h in wells containing 2 mL of DON (10 mg kg^−1^), ENB (10 mg kg^−1^), or a combination of the two mycotoxins (DON 10 mg kg^−1^ + ENB 10 mg kg^−1^). MDA content was analyzed immediately after the treatment. In all experiments, for each test, the same volume and concentration of DMSO was used as a control. Approximately 0.1 g (fresh weight, FW) of plant material (leaf or root) was homogenized with 1 mL of 0.1% trichloroacetic acid (TCA) (Sigma-Aldrich) and centrifuged at 10,000× *g* for 20 min. After centrifugation, 0.5 mL of supernatant was mixed with 1 mL of 0.5% TBA (Sigma-Aldrich) in 20% TCA. The mixture was incubated in boiling water for 30 min and then immediately cooled in an ice bath. Samples were centrifuged at 10,000× *g* for 10 min, and the absorbance was read at 532 nm and adjusted for nonspecific absorbance at 600 nm. MDA content was calculated by using an extinction coefficient of 155 (mM^−1^ cm^−1^) [78]. Experiments were replicated three times and, for each experiment, six samples/treatment were analyzed.

### 5.6. Fungal Material, Mycotoxins, and In Vitro Growth Analysis of FG and FA

Strains of FG (FG8) and FA (82DW09), previously characterized [26,44] and selected for their virulence on bread wheat seedlings and heads, were used to perform this test. Both strains were kept for seven days at 22 °C in the dark on plates containing potato dextrose agar (PDA) (Biolife Italiana, Monza, Milan, Italy). Additionally, for this experiment, the pure compounds ENB (BioAustralis) and DON (Sigma-Aldrich) were used. Both mycotoxins were solubilized in sterile and deionized water, aliquoted, and stored at −20 °C until use. To evaluate the effects of mycotoxins—applied alone or in combination—on fungal growth, one mycelium plug (0.5 cm diameter) of FG or FA was taken from the edge of a colony and placed in the middle of the plates containing PDA, to which different concentrations (10, 50, or 100 mg kg^−1^) of DON, ENB, or DON+ENB mixture had been previously added. The development of the fungus was assessed for a total of four days by measuring the area of the colony at 24, 48, 72, and 96 h using ImageJ. The same procedure was performed on control samples grown on PDA free of the mycotoxins. Seven replicates were performed for each concentration of the mycotoxins and their controls. Three independent experiments were conducted.

### 5.7. Statistical Analysis

Completely randomized designs were performed for all of the experiments. The data reported were tested by one-way ANOVA, followed by Tukey’s HSD test for the analysis of statistically significant differences (*p* ≤ 0.01).

## Figures and Tables

**Figure 1 toxins-13-00728-f001:**
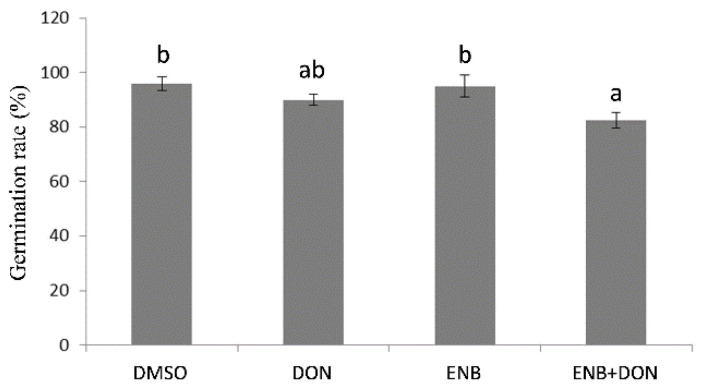
Germination of bread wheat seeds in response to single or combined deoxynivalenol (DON) and enniatin B (ENB) applications. Germination rate (%) was quantified in four independent experiments using 10 seeds per treatment in each of the experimental trials. In all experiments, for each test, the same volume and concentration of dimethyl sulfoxide (DMSO) was used as a control. Data represent the mean ± standard error. Different letters indicate significant differences in mean values (*p* ≤ 0.01, one-way ANOVA, Tukey’s HSD test).

**Figure 2 toxins-13-00728-f002:**
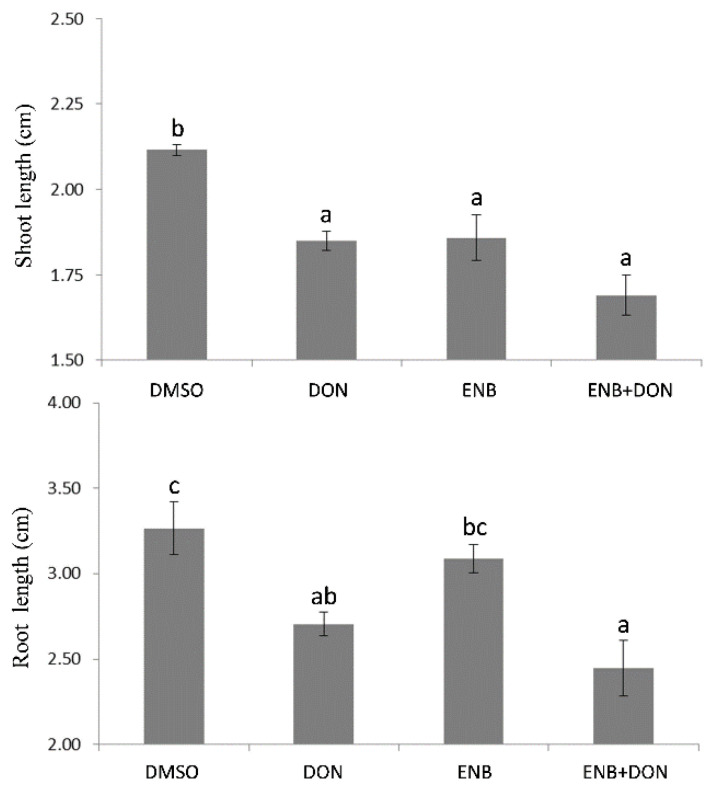
Effects of single or combined DON and ENB applications on bread wheat growth. In all experiments, for each test, the same volume and concentration of DMSO was used as a control. Data represent the mean ± standard error of four independent experiments. For each experiment, 10 replicates per treatment were analyzed. Different letters indicate significant differences in mean values (*p* ≤ 0.01, one-way ANOVA, Tukey’s HSD test).

**Figure 3 toxins-13-00728-f003:**
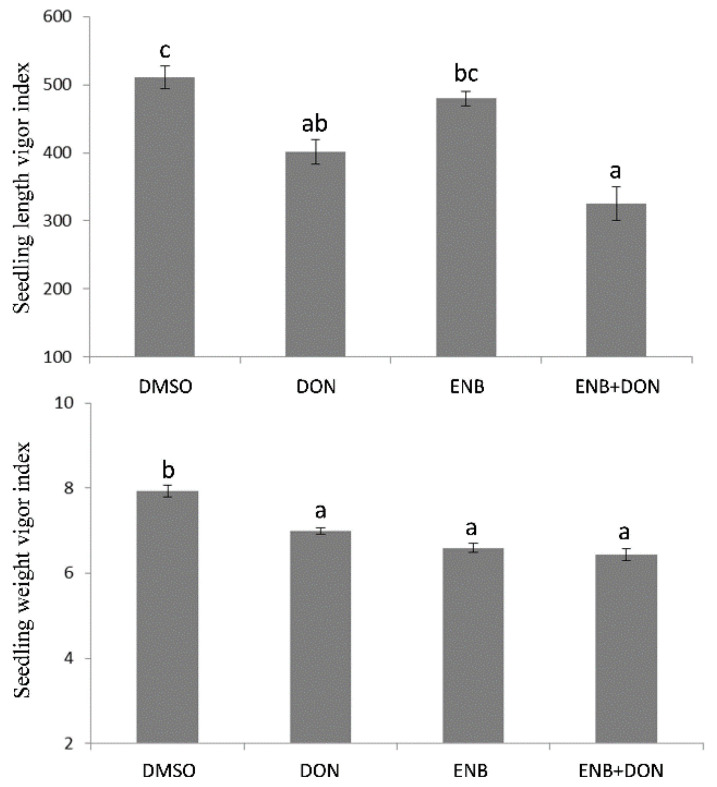
Vigor of bread wheat seedlings after single or combined DON and ENB applications. In all experiments, for each test, the same volume and concentration of DMSO was used as a control. Data represent the mean ± standard error of four independent experiments. For each experiment, 10 replicates per treatment were analyzed. Different letters indicate significant differences in mean values (*p* ≤ 0.01, one-way ANOVA, Tukey’s HSD test).

**Figure 4 toxins-13-00728-f004:**
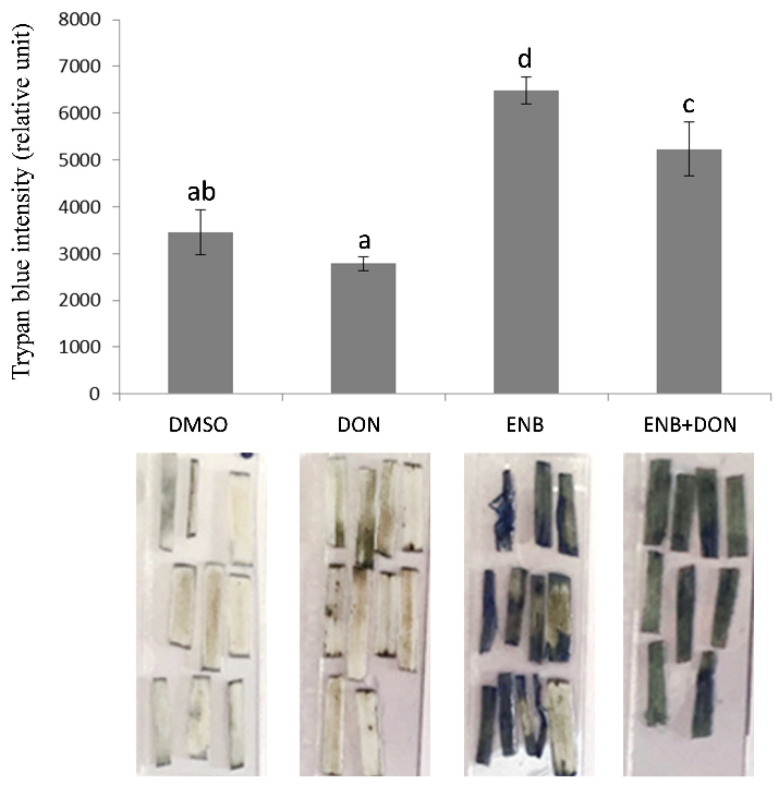
Quantification of cell death by trypan blue staining after single or combined DON and ENB applications. In all experiments, for each test, the same volume and concentration of DMSO was used as a control. Cell death was assessed by the intensity of trypan blue staining after seven days, as measured by ImageJ software. Data represent the mean ± standard error of three independent experiments with 8–10 leaf segments/treatment for each experiment. Different letters indicate significant differences in mean values (*p* ≤ 0.01, one-way ANOVA, Tukey’s HSD test). A representative image of stained plant tissues is shown below the histogram.

**Figure 5 toxins-13-00728-f005:**
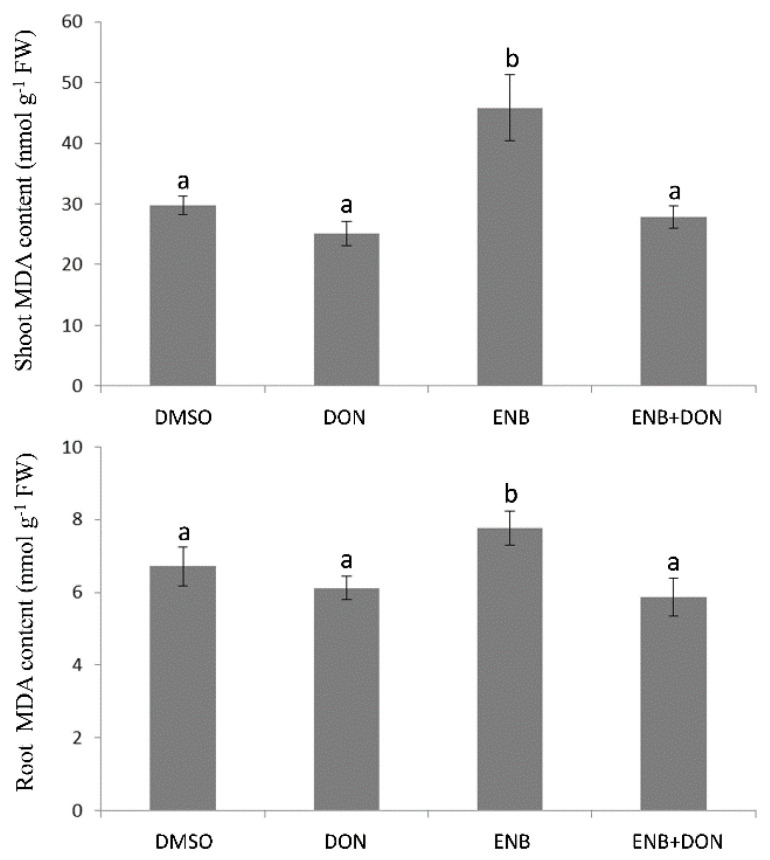
Malondialdehyde (MDA) content in bread wheat seedlings in response to single or combined DON and ENB applications. In all experiments, for each test, the same volume and concentration of DMSO was used as a control. Data represent the mean ± standard error of three independent experiments. For each experiment, six replicates per treatment were analyzed. Different letters indicate significant differences in mean values (*p* ≤ 0.01, one-way ANOVA, Tukey’s HSD test).

**Figure 6 toxins-13-00728-f006:**
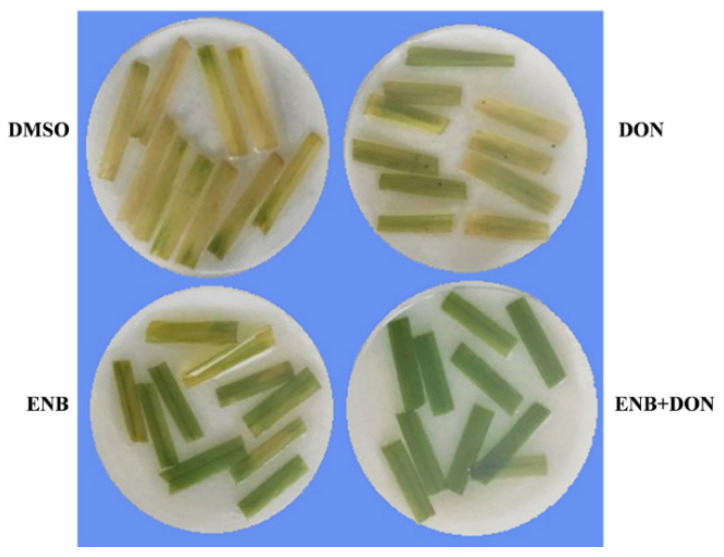
Change in pigment content after single or combined DON and ENB applications. In all experiments, for each test, the same volume and concentration of DMSO was used as a control. A representative image of one of the three performed independent experiments, each with 8–10 leaf segments/treatment, after seven days of incubation following mycotoxin application, is shown.

**Figure 7 toxins-13-00728-f007:**
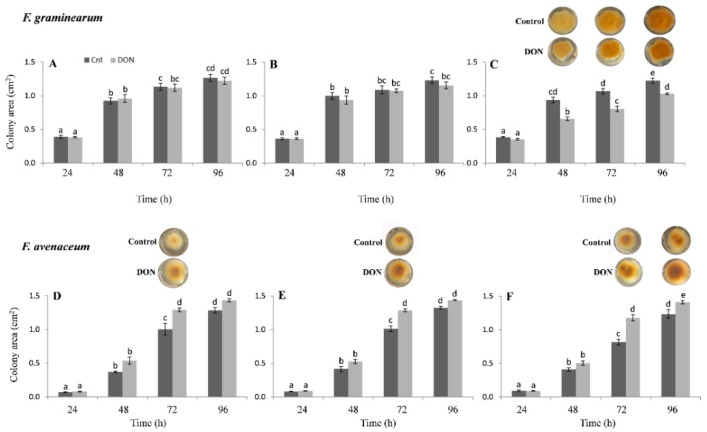
Effects of three DON concentrations on the in vitro growth of *Fusarium graminearum* (FG) and *Fusarium avenaceum* (FA). Seven replicates were performed for each concentration of DON and its controls (cnt). Data represent the mean ± standard error of three independent experiments. Different letters indicate significant differences in mean values (*p* ≤ 0.01, two-way ANOVA, Tukey’s HSD test). A representative image is provided above the histograms showing statistically significant differences between controls and treatments. (**A**,**D**): 10 mg kg^−1^; (**B**,**E**): 50 mg kg^−1^; (**C**,**F**): 100 mg kg^−1^.

**Figure 8 toxins-13-00728-f008:**
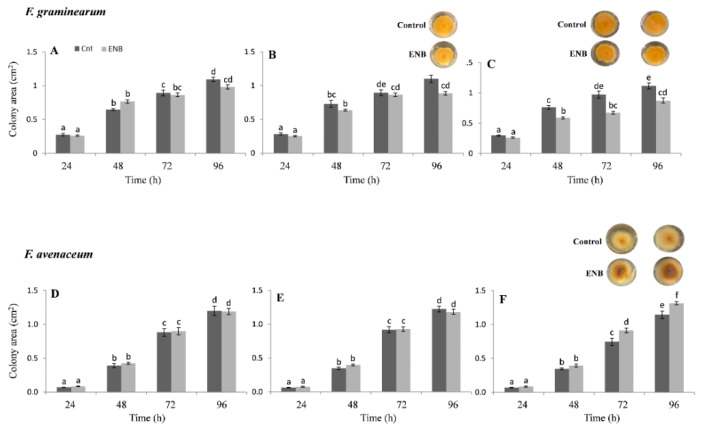
Effects of three ENB concentrations on the in vitro growth of FG and FA. Seven replicates were performed for each concentration of ENB and its controls. Data represent the mean ± standard error of three independent experiments. Different letters indicate significant differences in mean values (*p* ≤ 0.01, two-way ANOVA, Tukey’s HSD test). A representative image is provided above the histograms showing statistically significant differences between controls and treatments. (**A**,**D**): 10 mg kg^−1^; (**B**,**E**): 50 mg kg^−1^; (**C**,**F**): 100 mg kg^−1^.

**Figure 9 toxins-13-00728-f009:**
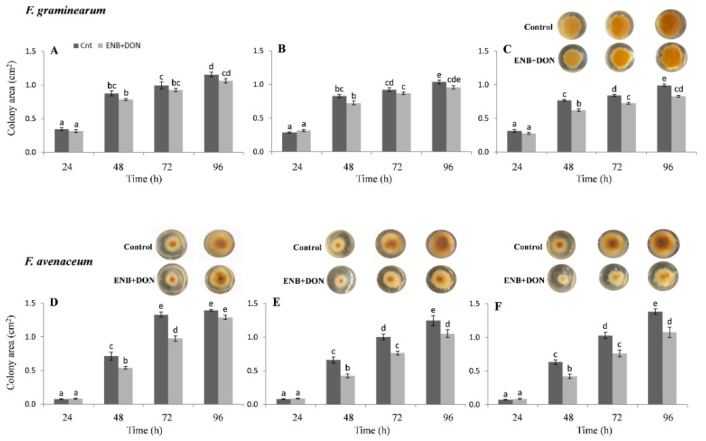
Effects of three concentrations of ENB+DON on the in vitro growth of FG and FA. Seven replicates were performed for each concentration of ENB+DON and their controls. Data represent the mean ± standard error of three independent experiments. Different letters indicate significant differences in mean values (*p* ≤ 0.01, two-way ANOVA, Tukey’s HSD test). A representative image is provided above the histograms showing statistically significant differences between controls and treatments. (**A**,**D**): 10 mg kg^−1^; (**B**,**E**): 50 mg kg^−1^; (**C**,**F**): 100 mg kg^−1^.

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
