# Peer review of "Enniatin B and Deoxynivalenol Activity on Bread Wheat and on Fusarium Species Development"

_toxins, 2021, doi:10.3390/toxins13100728_

Round 1

Reviewer 1 Report

This is an interesting manuscript about the effects of DON and ENB on physiology of wheat plants and their effect on F. graminearum and F. avenaceum growth in cultures. This is an important topic and one which lacks research progress in the literature. Overall, this study is well designed and I believe that it is a good candidate for being published in the Toxins journal. A few issues, however, need to be addressed. The manuscript requires some text editing; few specific examples are given below. A strengthened discussion section would be needed for acceptance. In particular, I would  like to see the authors elaborate more and  their opinion on the potential reasons of observed mycotoxin effects on leaf pigments, particularly that of ENB.

Specific comments:

Figure quality is rather low and difficult to read in the manuscript I received for review.

Fig. 1 – letters depicting statistical significance are missing.

Line 105-108: Indicate that synergistic effect of the two mycotoxins on inhibition of seedling development was significant ‘in comparison to the untreated control’; also, modify the sentence regarding root level effects as it seems obvious that the main ENB+DON effects are coming from DON and not from ENB. Same for vigor index, the difference was significant compared to the untreated control.

Line 121-122: modify the sentence to improve clarity; what is meant by ‘DON slightly reduced it’?.

Line 123: The combination of the two mycotoxins caused a significant increase in cell death.. indicate in comparison to which treatments.

Line 127:  However, a greater increase (+54%) in shoot… Why however, and add to the sentence shoot and root MDA content for clarity.

Line 181: Subtitle 2.4 – since statistics was done across treatments, add to the text where appropriate ‘in comparison to the corresponding control’ e.g. ‘Compared to the corresponding control, the treatment with DON did not induce significant changes on FG development at the lowest mycotoxin concentrations (10 and 50 mg kg-1), but it significantly reduced (-30%) colony growth after 48 h at the concentration 100 mg kg-1 (Figure 7A-C), etc…

Line 227: modify the sentence to improve clarity e.g. suggestion ‘Simultaneous exposure of FG to both mycotoxins (ENB+DON) resulted in general decrease of the fungal growth at all mycotoxin rates, however, in comparison to the corresponding control this difference was statistically significant only at 100 mg kg-1 mycotoxin levels’.

Line 230: change added with amended.

Lines 262 , 298, 329,365: Change Reference [52] with authors names.

Line 346: change determined with resulted.

Line 351: Please provide explanation what is meant by ‘certain “positive” effect’. The current text provides insufficient detail.

Author Response

Dear reviewer, 

We are very grateful for your valuable revision that certainly improves the quality of the paper. Please find the detailed response in the attached file. 

Kind regards. 

Reviewer 2 Report

The research data presented in the manuscript are relevant to the study of the role of two important mycotoxins (enniatin B and deoxynivalenol) produced by two different Fusarium species in their phytotoxicity in soft wheat and competition between fungi occupying the same ecological niche.

However, the article contains a number of inaccuracies that need to be explained:

How many samples did you analyses?

Did you check concentration of ENB, DON and (DON+ENB) in seeds after incubation in solution?

If not, it is difficult understand the practical relevance of study, because the concentration of mycotoxins in seeds can be different from its level in solution.

Line 91-96, 134-137, 142-144, 150-154, 160-163, 175-179, 206-211, 217-222, 234- 239 – I think methodology of study should be in parts of materials and methods. You need to add notes to explain short abbreviation.

The discussion and conclusion should be improved.

Author Response

(The authors gave the same response as above.)

Round 2

Reviewer 2 Report

The authors have corrected manuscript according to my comments and suggestions.